# OpenReview forum: "Beyond Standardization – Putting the Normality in Normalization"
_ICLR.cc/2025/Conference — Submitted to ICLR 2025_

### Official Review · Reviewer_hs2Z · 2024-10-21

**Soundness:** 2
**Presentation:** 3
**Contribution:** 2
**Rating:** 3
**Confidence:** 4

**Summary:**

This paper proposes a new type of normalization layer for neural networks, to encourage the pre-activation distributions to be Gaussian. This is motivated by several information-theory arguments such as increasing robustness to noise. The layer is composed of standard normalization, a power transform (in which a single power parameter is determined by approximately maximizing the Gaussian likelihood), centralization, and the addition of Gaussian noise. The benefits of these layers are examined empirically.

**Strengths:**

1. The idea is interesting, original, novel, and with a reasonable motivation.

2. The proposed layer does seem to improve generalization performance in all cases it was tested on, compared to the existing normalization layers (batch-norm, layer-norm, group-norm, and instance-norm).

3. The experiments convincingly demonstrate the improvements in normality and noise robustness after using the new normalization layer.

4. The effect of various quantities, such as width, depth, and minibatch size is examined.

5. The presentation is clear and informative.

**Weaknesses:**

1. The main issue of this paper is the scale of the experiments. For this type of paper, the bare minimum is an Imagenet experiment and possibly also some Language model fine-tuning. However, this paper stops at the scale of tiny Imagenet and CIFAR. This is crucial since many methods work well on such small datasets but not in Imagenet (for example, weight normalization).

2. Even the existing results are compared to suspiciously low accuracy baselines. For example, ResNet18 in CIFAR10 with standard BN achieves 88.89% test accuracy, while the first GitHub repo I found on Google search achieves 93.02 accuracy (https://github.com/kuangliu/pytorch-cifar), which is better than the 90.41% accuracy reported using the proposed BNN method. This is important, since in many cases using better baselines can cause the improvement to narrow or even disappear. Ideally, for each model and dataset, we need a baseline near the current state-of-the-art and show the new method improves. The most convincing thing would be to show the state-of-the-art is improved for a dataset (using the best model), but I acknowledge this may require too large resources.

3. Missing ablation studies: how much each part of the proposed layer is contributing to the improvement? e.g. is the power transform more important than the added noise? How sensitive are we to the $\xi$  parameter? This is important, since if most of the benefits come from the added Gaussian noise, this takes away some of the novelty of the method, as this is somewhat similar to Gaussian dropout, which has already been suggested before (“Fast Droput Training” ICML 2013, “Variational Dropout and the Local Reparameterization Trick” NeurIPS 2015).

Minor points:

line 150: “No additional parameters” title is misleading, even though the paragraph says no additional learnable parameters since the $\xi$  is a free parameter (but not a learned parameter)

Table 2: some are the test accuracy results are extremely low (e.g. 66.56% on CIFAR10), probably because ViTs don't work well in small datasets. If we must test layer norm on small datasets, then I would use an architecture with more reasonable performance, such as Convnext.

Motivation section: it is a bit strange that this section appears toward the end. It's more common to write this at the beginning.

line 499: I'm not sure what the line “Seldom has the question of precisely what distribution a deep learning model should use to effectively encode its representations” means. I think this has been investigated in many different contexts, for example, the information bottleneck papers and the quantization literature (where some distributions are easier to quantize than others).

**Questions:**

See weakness section.

---

> ### Author Response · Authors · 2024-11-22
>
> To our dear Reviewer,
>
> We address each of your comments below:
>
> >
> >"The main issue of this paper is the scale of the experiments. For this type of paper, the bare minimum is an Imagenet experiment and possibly also some Language model fine-tuning. However, this paper stops at the scale of tiny Imagenet and CIFAR. This is crucial since many methods work well on such small datasets but not in Imagenet (for example, weight normalization)."
> >
>
> We completely understand your perspective here, i.e. that larger scale experiments would provide further evidence that the method works well, citing ImageNet as an example.
>
> We would kindly like to point out that we do have ImageNet experiments in Table 2, in the form of the ImageNet100 experiments, which convincingly demonstrated the superior performance of the vision transformer (ViT) trained with layer normality normalization (LNN) compared to the ViT trained with layer normalization (LN). This portion of Table 2 reads:
>
> |Dataset|LN|LNN|
> |----------|----------|----------|
> |ImageNet100 Top1|50.78 $\pm$ 0.33|**62.39 $\pm$ 0.68**|
> |ImageNet100 Top5|75.45 $\pm$ 0.50|**84.03 $\pm$ 0.42**|
>
> Furthermore, because we have run all of our experiments with $M=6$ random initializations for the model parameters, and provide their final mean performance across the $M=6$ models, we provide evidence for a high degree of confidence/precision in our results. We ran ($M=6 \times 2$) ImageNet100 experiments – one set of $M=6$ experiments for models trained with LNN, and another $M=6$ experiments for models trained with LN – and the results we obtained were strong. We decided that rather than dispatching another $12$ experiments for ImageNet altogether, that demonstrating strong performance across many dataset & model combinations would be most informative, again given how strong the performance was on ImageNet100.
>
> Finally, please note that as described in Appendix C.2, for each of the $M=6$ ImageNet100 experiments, we subsampled a different set of 100 classes at random (whilst ensuring that these subsampled classes are precisely the same ones in the experiments with LNN vs. LN, for fair comparison). By doing this, rather than re-using the same 100 classes for each ImageNet100 experiment, we demonstrate even greater precision in our aggregated results for the ImageNet dataset.
>
> We also considered fine-tuning language models. However, after preliminary investigation, we found that substituting a normalization layer after a model has already been trained, makes the comparison to the substituted normalization unfair, and often does not work well. To summarize, substituting the normalization layer of a model trained with one normalization layer (ex: LN) with another normalization layer (ex: LNN) midway through training, does not provide a fair setting for the second normalization layer, because this second normalization layer does something intrinsically different than the first one, which the model was trained with up to that point. To verify this was not a characteristic unique to LNN, we observed the same behavior when we trained language models from scratch using LNN, then attempted to fine-tune using LN – again the performance suffers, because the model is being fine-tuned using a different normalization layer. Thus regardless of whether one starts with LN/LNN, then changes to LNN/LN for fine-tuning, the setup does not provide a fair chance for the second normalization layer to perform as well as it would, if the model had been trained with said normalization layer from scratch. We believe the fair comparison would have been to train a large language model from scratch (random initialization) using LNN – however due to the resources this would require, and because our experimental results were strong altogether, we decided that dedicating a large amount of resources to training a large language model from scratch would take away from our ability to run our other experiments.
>
> We sincerely believe the existing experiments provide sufficient and convincing evidence, including on the large-scale ImageNet dataset, that normality normalization is a highly effective normalization layer across the board.

---

> ### Author Response · Authors · 2024-11-22
>
> >
> >"Even the existing results are compared to suspiciously low accuracy baselines. For example, ResNet18 in CIFAR10 with standard BN achieves 88.89% test accuracy, while the first GitHub repo I found on Google search achieves 93.02 accuracy (https://github.com/kuangliu/pytorch-cifar), which is better than the 90.41% accuracy reported using the proposed BNN method. This is important, since in many cases using better baselines can cause the improvement to narrow or even disappear. Ideally, for each model and dataset, we need a baseline near the current state-of-the-art and show the new method improves. The most convincing thing would be to show the state-of-the-art is improved for a dataset (using the best model), but I acknowledge this may require too large resources."
> >
>
> We completely understand your perspective here, and would like to clarify why the baseline performances are lower: we did not use augmentations. The repository you refer to, does use augmentations, which is the reason for the higher performance. We chose not to run experiments with augmentations, in order to control for the effect of the normalization layer alone, without conflating this effect with other extraneous factors.
>
> We have now however also run an experiment to verify the performance of ResNet18 x CIFAR10 using BNN, when using precisely the same data augmentations used in the repository you linked to. These were `transforms.RandomCrop(32, padding=4)` and `transforms.RandomHorizontalFlip()`. Across $M=6$ runs, we obtained a mean performance of 94.93% $\pm$ 0.05, which surpasses the performance listed in the repository.

---

> ### Author Response · Authors · 2024-11-22
>
> >
> >"Missing ablation studies: how much each part of the proposed layer is contributing to the improvement? e.g. is the power transform more important than the added noise? How sensitive are we to the $\xi$ parameter?"
> >
>
> We address this comment through Appendix D.2 Controlling for the Power Transform and the Additive Noise, where through Figure 8 we conducted additional experiments demonstrating the effect each component of the normalization layer has. We controlled for the effect of the power transform by setting $\xi$ to $0$; these models are denoted by "BNN w/o noise". The results demonstrate a clear benefit from both components of the normalization layer; the power transform, and the additive Gaussian noise with scaling.
>
> Regarding the sensitivity to the free parameter $\xi$, we set this to a single value for the two types of architectures we used (ResNet/WideResNet and ViT). We demonstrated that with this single value, models performed well across the board; despite changes in dataset, architecture size (depth/width), and minibatch size.
>
> To address the question of how this value for $\xi$ was chosen, as we describe in Appendix Subsections C.1 and C.2, these were chosen solely using preliminary experiments which aimed to evaluate, at what point further increases to $\xi$ led to unstable training behaviors. Given we used a consistent value for $\xi$ across our experiments for the ResNet/WideResNet and ViT architectures, this shows the effectiveness of the method was not sensitive to the value of $\xi$.
>
> Finally, we would like to highlight that our contribution is dual in nature – both the power transform component, and the additive gaussian noise with scaling component, are distinct and novel contributions. Furthermore, due to the motivation we explored in the paper, in particular through the mutual information game, these two contributions act to supplement and reinforce each other.

---

> ### Author Response · Authors · 2024-11-22
>
> >
> >"This is important, since if most of the benefits come from the added Gaussian noise, this takes away some of the novelty of the method, as this is somewhat similar to Gaussian dropout, which has already been suggested before (“Fast Droput Training” ICML 2013, “Variational Dropout and the Local Reparameterization Trick” NeurIPS 2015)."
> >
>
> We are excited that you brought up a comparison to Gaussian dropout, as it presents an opportunity to contrast and compare the two methods. First, Appendix D.1 Other Noise-Based Techniques now compares and contrasts our proposed method with Gaussian dropout, over several retention probabilities $p$, as shown in Figure 7. Here we demonstrate that our proposed noising method works better. We also show our method works best when $s$ is set according to the minibatch statistics, i.e. not as a fixed constant, which adds further novelty and value to the method.
>
> Furthermore, as we explore in the text of Appendix D.1 Other Noise-Based Techniques, there is a significant difference between works applying Gaussian dropout, and the present work which uses additive Gaussian noise with scaling. Gaussian dropout scales activations multiplicatively, which has the following subtle but significant consequence: the effect and scale of the noise is incorporated directly during gradient descent via backpropagation – this boils down to the fact that multiplicative operations carry over when taking gradients. In contrast, the additive gaussian noise is not directly incorporated into the gradient descent updates during backpropagation, because additive effects are eliminated when taking gradients. In this sense, the noise from additive Gaussian noise is "confusable", because the backward pass accounts for a different activation value than what was realized during the forward pass. This implies that models which can successfully be trained with this additive Gaussian noise, should be more robust, and have better generalization – which our experiments demonstrate.

---

> ### Author Response · Authors · 2024-11-22
>
> >
> >"line 150: “No additional parameters” title is misleading, even though the paragraph says no additional learnable parameters since the $\xi$ is a free parameter (but not a learned parameter)"
> >
>
> Thank you for pointing this out - we have modified the title of this paragraph to "No Additional Learned Parameters".
>
> >
> > "Table 2: some are the test accuracy results are extremely low (e.g. 66.56% on CIFAR10), probably because ViTs don't work well in small datasets. If we must test layer norm on small datasets, then I would use an architecture with more reasonable performance, such as Convnext."
> >
>
> Thank you for highlighting this point – we agree that demonstrating our proposed normalization layer improves in performance with commonly used techniques, is important. Therefore we have added a new set of experiments in Appendix D.3 Experiments with Data Augmentations via Table 4, which demonstrates the improvement in performance that can be leveraged by employing commonly used techniques such as data augmentations, whilst still demonstrating that the models trained with LNN perform better than those trained with LN. Table 4 is given as follows:
>
> |Dataset|LN|LNN|
> |----------|----------|----------|
> |SVHN|94.46 $\pm$ 0.33|**95.94 $\pm$ 0.18**|
> |CIFAR10|73.71 $\pm$ 0.42|**75.47 $\pm$ 0.49**|
> |CIFAR100|49.56 $\pm$ 0.42|**52.89 $\pm$ 0.51**|
> |Food101|55.43 $\pm$ 0.57|**63.04 $\pm$ 0.72**|
>
> This demonstrates a significant improvement from the results in Table 2, which is facilitated through the commonly used technique of data augmentations. This demonstrates that the improvements for models trained with LNN continue to scale with the use of such techniques.
>
> >
> > "Motivation section: it is a bit strange that this section appears toward the end. It's more common to write this at the beginning."
> >
>
> We have moved the motivation section, so that it now comes directly after the introduction. We agree that the logic of the paper flows better with the motivation in Section 2.
>
> >
> > "line 499: I'm not sure what the line “Seldom has the question of precisely what distribution a deep learning model should use to effectively encode its representations” means. I think this has been investigated in many different contexts, for example, the information bottleneck papers and the quantization literature (where some distributions are easier to quantize than others)."
> >
>
> We agree with this point, and have modified the sentence to a) reflect specifically that we are talking about the activations in neural network layers, and b) that seldom has an exact prescription for what this distribution should be, and how to achieve it in a practical manner, been provided before. The sentence now reads "Seldom has a prescription for precisely what distribution a deep learning model should use to effectively encode its activations, and exactly how this can be achieved, been investigated.".

---

> ### Author Response · Authors · 2024-11-22
>
> Furthermore, we have made the following very valuable additions to the paper:
> 1. We have added a new section Appendix D.4 Effect of Degree of Gaussianization, which explores how the extent of the gaussianization relates to model performance via Figure 10, demonstrating that increasing gaussianity does improve performance,
> 1. We have added a new section Appendix D.8 Uncorrelatedness, Joint Normality, and Independence Between Features, which demonstrates via Figure 14 the increased joint gaussianity normality normalization imbues, the resulting reduced correlation between channels of the same layer, and the increased extent of independence between channels of the same layer, the latter of which has previously been shown to be beneficial in neural networks, as we describe in Subsection 2.3,
> 1. We have a new motivation in Subsection 2.3 Maximally Independent Representations, which explores feature correlation, joint normality, and independence, between channels in the context of gaussianization, citing why increased independence can be valuable in learning models,
> 1. We have added a new section Appendix D.7 Normality at Initialization, demonstrating via Figure 13 that at initialization, both BatchNormalNorm and BatchNorm exhibit gaussianity; but that via Figure 5, only BatchNormalNorm enforces and maintains this gaussianity through training,
> 1. We have added a new section Appendix D.5 Training Convergence, demonstrating via Figure 11 that the general trends in training and validation curves remain similar when using normality normalization. This is valuable because it suggests the understanding deep learning practitioners have obtained for training models with conventional normalization layers, remains applicable when augmenting those normalization layers using normality normalization,
> 1. We have added a new paragraph in Section 6 Related Work & Future Directions: Gaussianization, regarding other gaussianization techniques which may be of interest for future work,
> 1. In the introduction we have added further motivation for gaussianity in paragraph 3, through the perspective of neural networks as gaussian processes,
> 1. We have changed the use of the term standardization, to align more closely with the deep learning literature, which conventionally uses the term normalization. This was done to avoid the possibility of confusing the reader – for this reason we have also changed the paper title,
> 1. We have made several improvements throughout the text.
>
> We'd really like to thank you for your time and consideration – your review has helped further strengthen the work.
>
> We have sincerely made every attempt to comprehensively and concretely address each of your comments; through the added experiments, the additional analyses, and the refinements made to the paper. Additionally, we have made several further improvements to the work, which we listed here.
>
> Given this, we sincerely ask that you consider increasing your score.

---

> > ### Comment · Reviewer_hs2Z · 2024-11-24
> > **Response to Rebuttal**
> >
> > I have read all the reviews and responses. I thank the authors for their laudable efforts, which have addressed most of my concerns. However, my two main concerns remain:
> > 1. The small scale of the experiments. Here I should clarify that by 'scale', I didn't mean the size of the images, but the number of data points and the task's difficulty. So ImageNet100, as it has 10^2 classes and 10^5 samples, does not improve the scale here in comparison to the other tasks that have a similar scale (TinyImageNet, CIFAR100) or smaller scale. I understand the authors want to average on multiple seeds, but I find the lack of any Full ImageNet experiment suspicious (as it should not be so difficult). Given the lack of time until the end of the rebuttal period, even a single full ImageNet experiment here (M=1) showing a significant improvement would be enough for me to raise the score.
> > 2. The low accuracy of the baselines. I thank the authors for the new results with data augmentations on ResNet 18 and CIFAR10, and the new Table 4. The ResNet18 result is promising, but the results in Table 4 still make the impression that most of the baselines in the paper are still very low. So I'm still uneasy that most of the improvement will 'wash away' as we get closer to the state-of-the-art, as is often the case in many methods.

---

> > > ### Author Response · Authors · 2024-11-26
> > >
> > > Dear Reviewer,
> > >
> > > We appreciate your transparency regarding experimental results on ImageNet. Earlier on (even before your most recent comment), we dispatched precisely such an experiment involving ResNet50 models trained with either BatchNormalNorm and BatchNorm, for which we now share the results for: Appendix D.3 Experiments with Data Augmentations has been updated with the results of this experiment.
> > >
> > > The experiment demonstrates an improvement for the ResNet50 model trained with BatchNormalNorm. Furthermore, this experiment was run with $\xi=0$, i.e. using BatchNormalNorm without noise (BNN w/o noise), and still demonstrated an improvement over BatchNorm (BN). This further acts to control for the effect of the power transform alone; which is also a point of inquiry you had raised in your original review.
> > >
> > > This most recent evidence we provide comes in addition to our previous rebuttal in response to your review, for which you shared had addressed most of your concerns. Altogether the results and analyses demonstrate that normality normalization is a highly effective normalization layer across a wide range of dataset scales; this is in addition to the many useful and interesting properties of normality normalization, which we have explored throughout the paper, and have furthermore demonstrated during the rebuttal period.
> > >
> > > In light of this, we would be highly appreciative if you could take a moment to consider increasing the score for our submission.

---

> > > > ### Comment · Reviewer_hs2Z · 2024-11-26
> > > > **Response**
> > > >
> > > > I thank the authors for running the experiment. Unfortunately, these results do not yet convince me that the BNN method is practically useful beyond small-scale tasks:
> > > > 1. Mainly, the improvement is not very significant (0.13% in Top5 and 0.34% in Top1), especially given this is a single seed and the following issues.
> > > > 2. The baseline accuracy (71.6%) is still suspiciously low (typically, ResNet50 has an accuracy > 75%).
> > > > 3. The results are only shown for $\xi=0$, while the recommended value (used throughout the paper) is $\xi=0.4$. This also seems strange.
> > > >
> > > > Therefore, I still cannot recommend acceptance. Many papers significantly improve small-scale tasks but fail to improve significantly on a larger scale (e.g., ImageNet). This paper must convincingly demonstrate this scalability or this method will not be widely adopted (even if it works well).

---

> ### Author Response · Authors · 2024-12-03
>
> Dear Reviewer,
>
> After your earlier review comment, to which we are replying to in the present comment, we sought to further address the concern you quoted regarding the low accuracy of the baselines, and to demonstrate that normality normalization continues to scale in performance with the use of additional techniques for improving generalization performance. Here we convincingly address these items through the experiments we describe next.
>
> We first describe the training details and the configurations for the experiments we ran. We then present the results. Finally, we comment on the results and the conclusions which can be drawn from them.
>
> We used the same ViT model configuration, and the same optimizer setup, as in our other experiments; these are detailed in Appendix Subsection C.2. The present experiments, however, differed in the following ways. We used a training minibatch size of 128 throughout the experiments. For the CIFAR10 and CIFAR100 datasets, we trained models for 900 epochs, and for the Food101 dataset, we trained models for 300 epochs. We employed a learning rate warmup strategy, where the learning rate was linearly increased from a fraction of its base value to the full learning rate. This was implemented using pytorch's LinearLR scheduler, using a start_factor of 0.1 and total_iters of 10. (documentation: pytorch.org/docs/stable/generated/torch.optim.lr_scheduler.LinearLR.html). After the warmup phase, a cyclic learning rate schedule based on cosine annealing with periodic restarts was employed. This was implemented using pytorch's CosineAnnealingWarmRestarts scheduler, with T_0=50, T_mult=2, eta_min=1e-6. (documentation: pytorch.org/docs/stable/generated/torch.optim.lr_scheduler.CosineAnnealingWarmRestarts.html). We used the same data augmentations listed in Appendix Subsection D.3.
>
> Given the time constraints of the rebuttal period, we focused on conducting experiments which would most conclusively determine whether models trained with additional techniques for improving generalization performance, would continue to yield the most benefit when using normality normalization. Therefore, we did not consider additional experiments on the SVHN dataset, because the performance levels obtained in Table 4 were already very strong for this dataset. To the extent that it was possible, we also employed multiple seeds for training models from differing random initializations for the model parameters. Specifically, for the Food101 dataset, $M=6$ models were trained, and for the CIFAR10 dataset, $M=4$ models were trained; we sought to further substantiate our findings through the multiple random seeds. We also, however, ran an additional experiment using the CIFAR100 dataset, to provide further coverage across multiple datasets, and to help further make the results conclusive. As seen from the results below, they all point conclusively to the continued improvement in performance for models trained with normality normalization, when using additional techniques for improving generalization performance.
>
> |Dataset|LN|LNN|
> |----------|----------|----------|
> |CIFAR10 (M=4)|80.42 $\pm$ 0.29|**82.97 $\pm$ 0.14**|
> |CIFAR100 (M=1)|53.18|**58.47**|
> |Food101 (M=6)|61.61 $\pm$ 0.31|**69.11 $\pm$ 0.20**|
>
> These results provide strong evidence that models trained with normality normalization continue to improve with the use of additional techniques for improving generalization performance, and that they continue to outperform models trained with other normalization layers.
>
> Furthermore, these results are a significant improvement to the results in Table 4. The gains (LNN) also occur in comparison to higher baseline levels of performance (LN), further addressing your comment.

---

> ### Author Response · Authors · 2024-12-03
>
> We have throughout the work aimed to comprehensively demonstrate, that normality normalization is a highly performant normalization layer. We have done so through extensive experiments, originally with a focus on controlling for various factors of variation. We have here furthermore demonstrated that normality normalization continues to scale and outperform other normalization layers, when additional techniques for improving generalization performance are employed.
>
> Finally, we believe it is quite appropriate to briefly reflect on the present review & rebuttal setting. In our original experiments we had sought to demonstrate normality normalization's effectiveness across a multitude of factors, including a wide array of commonly used model and dataset combinations, common factors of variation such as model width, depth, and training minibatch size, and to demonstrate its suitability across various normalization layers. We had furthermore done this with an emphasis on presenting our results with precision and confidence, as can be seen by our use of multiple random seeds. We demonstrated both the strong performance of normality normalization, in addition to having explored numerous useful and interesting properties of the proposed normalization layer. Your review, and our subsequent responses, have helped further substantiate that the proposed normalization layer is performant, and that it also scales with the use of additional techniques for improving generalization performance. We ultimately believe this process will have served to further encourage & expedite the adoption of the normalization layer, and to draw further interest towards exploring its many interesting properties – and this is something to be grateful for.
>
> Thus we ask here that you consider, any, increase to your score for our submission.

---

### Official Review · Reviewer_BP3W · 2024-10-25

**Soundness:** 3
**Presentation:** 3
**Contribution:** 3
**Rating:** 5
**Confidence:** 4

**Summary:**

The authors introduce normality normalization, a variant of batch normalization. Instead of just normalizing the first two moments, the authors normalize the distribution to be approximately normal. The normality normalization relies on the so-called power transformation, which can be approximated with an iterative method. The authors demonstrate consistent gains for many small-scale computer vision datasets and models when using normality normalization instead of batch/layer-norm. They also provide a whole section dedicated to motivation and the relationship to information theory.

**Strengths:**

- Normalization is common and impactful.
- They authors consider multiple models, datasets and normalization baselines and show consistent improvements.

**Weaknesses:**

- The experimental parts are relatively small-scale. Would be interesting to train e.g. a small GPT-2 style model.
- Most people might find section 5 (motivation), to not be very relevant. The good empirical results is all the motivation was is needed. :)
- The motivation behind the hyperparameter selection is not clear.

**Questions:**

- How was the hyperparameters selected for the experiments?
- Are you able to run a more large scale experiment?


# Update

I've reviewed the comments from the other reviewers. I note one mentioning that `The baseline accuracy (71.6%) is still suspiciously low (typically, ResNet50 has an accuracy > 75%).`. I am inclined to agree with this reviewer. I didn't catch this issue when reading the paper myself, but applaud the reviewer for his/her diligence. In light of this issue, I've decreased the score to a 5.

---

> ### Author Response · Authors · 2024-11-22
>
> To our dear Reviewer,
>
> We address each of your comments below:
>
> >
> >"The experimental parts are relatively small-scale. Would be interesting to train e.g. a small GPT-2 style model." and "Are you able to run a more large scale experiment?"
> >
>
> We would kindly like to point out that we did have experiments for the large-scale ImageNet dataset in Table 2, in the form of ImageNet100 experiments, which convincingly demonstrated the superior performance of the vision transformer (ViT) trained with layer normality normalization (LNN) compared to the ViT trained with layer normalization (LN). This segment of Table 2 is given as follows:
>
> |Dataset|LN|LNN|
> |----------|----------|----------|
> |ImageNet100 Top1|50.78 $\pm$ 0.33|**62.39 $\pm$ 0.68**|
> |ImageNet100 Top5|75.45 $\pm$ 0.50|**84.03 $\pm$ 0.42**|
>
> Furthermore, we have contrasted our approach across many factors of variation, such as dataset, model, normalization type, and other factors. Each result in the paper, numerical or graphical, represents the aggregate mean performance across $M=6$ models each of which had differing random seeds during training – this gives more weight to our results, by means of them being more precise. In general, we have taken this facet of the experiments – of comprehensiveness and precision in the reporting of our results – quite seriously, and we believe this is evidenced throughout the paper.

---

> ### Author Response · Authors · 2024-11-22
>
> >
> >"The motivation behind the hyperparameter selection is not clear." and
> >"How was the hyperparameters selected for the experiments?"
> >
>
> We provide a complete description of how hyperparameters were selected in Appendix Subsections C.1 and C.2. We investigated several hyperparameter configurations, including for the learning rate, learning rate scheduler, weight decay, and minibatch size, across all the models and found the presented configurations to generally work best across all of them.
>
> We further address this comment through Appendix D.2 Controlling for the Power Transform and the Additive Noise, where through Figure 8 we conducted additional experiments demonstrating the effect each component of the normalization layer has. We controlled for the effect of the power transform by setting $\xi$ to $0$; these models are denoted by "BNN w/o noise". The results demonstrate a clear benefit from both components of the normalization layer; the power transform, and the additive Gaussian noise with scaling.
>
> Regarding the sensitivity to the free parameter $\xi$, we set this to a single value for the two types of architectures we used (ResNet/WideResNet and ViT). We demonstrated that with this single value, models performed well across the board; despite changes in dataset, architecture size (depth/width), and minibatch size.
>
> To address the question of how this value for $\xi$ was chosen, as described in Appendix Subsections C.1 and C.2, these were chosen solely using preliminary experiments which aimed to evaluate, at what point further increases to $\xi$ led to unstable training behaviors. Given we used a consistent value for $\xi$ across our experiments for the ResNet/WideResNet and ViT architectures, this shows the effectiveness of the method was not sensitive to the value of $\xi$.

---

> ### Author Response · Authors · 2024-11-22
>
> Furthermore, we have made the following very valuable additions to the paper:
> 1. We have added a new section Appendix D.1 Other Noise-Based Techniques, where we investigate how differing noise techniques such as Gaussian dropout, compare to our proposed method of additive Gaussian noise with scaling, and for which the results are shown in Figure 7. We demonstrate that our proposed noising method works better. We also show our method works best when $s$ is set according to the minibatch statistics, i.e. not as a fixed constant, which adds further novelty and value to the method. The discussion contained in Appendix D.1 is also of interest.
> 1. We have added a new section Appendix D.4 Effect of Degree of Gaussianization, which explores how the extent of the gaussianization relates to model performance via Figure 10, demonstrating that increasing gaussianity does improve performance,
> 1. We have added a new section Appendix D.8 Uncorrelatedness, Joint Normality, and Independence Between Features, which demonstrates via Figure 14 the increased joint gaussianity normality normalization imbues, the resulting reduced correlation between channels of the same layer, and the increased extent of independence between channels of the same layer, the latter of which has previously been shown to be beneficial in neural networks, as we describe in Subsection 2.3,
> 1. We have a new motivation in Subsection 2.3 Maximally Independent Representations which explores feature correlation, joint normality, and independence, between channels in the context of gaussianization, citing why increased independence can be valuable in learning models,
> 1. We have added a new section Appendix D.3 Experiments with Data Augmentations, where via Table 4 we demonstrate that the improvements in models trained with LNN continue to scale when employing commonly used techniques such as data augmentations, whilst still demonstrating that the ViT models trained with LNN perform better than those trained with LN.
> 1. We have added a new section Appendix D.7 Normality at Initialization, demonstrating via Figure 13 that at initialization, both BatchNormalNorm and BatchNorm exhibit gaussianity; but that via Figure 5, only BatchNormalNorm enforces and maintains this gaussianity through training,
> 1. We have added a new section Appendix D.5 Training Convergence, demonstrating via Figure 11 that the general trends in training and validation curves remain similar when using normality normalization. This is valuable because it suggests the understanding deep learning practitioners have obtained for training models with conventional normalization layers, remains applicable when augmenting those normalization layers using normality normalization,
> 1. We have added a new paragraph in Section 6 Related Work & Future Directions: Gaussianization, regarding other gaussianization techniques which may be of interest for future work,
> 1. In the introduction we have added further motivation for gaussianity in paragraph 3, through the perspective of neural networks as gaussian processes,
> 1. We have changed the use of the term standardization, to align more closely with the deep learning literature, which conventionally uses the term normalization. This was done to avoid the possibility of confusing the reader -- for this reason we have also changed the paper title,
> 1. We have made several improvements throughout the text.
>
> We'd really like to thank you for your time and consideration – your review has helped further strengthen the work.
>
> We have sincerely made every attempt to comprehensively and concretely address each of your comments; through the added experiments, the additional analyses, and the refinements made to the paper. Additionally, we have made several further improvements to the work, which we listed here.
>
> Given this, we sincerely ask that you consider increasing your score.

---

> > ### Author Response · Authors · 2024-11-26
> >
> > Dear Reviewer,
> >
> > We would like to follow-up on our previous response to your review, in which we carefully addressed your comments, and submitted a revised version of our manuscript.
> >
> > Furthermore, we have added new experimental results on the large-scale ImageNet dataset in Appendix D.3 Experiments with Data Augmentations. This serves to further address your earlier comment about running a larger scale experiment. It also serves to further address your comment regarding the motivation and effect of the hyperparameters, for example as it pertains to $\xi$ in this context, since $\xi$ was set to $0$ (BNN w/o noise) and still led to an improvement in performance over BN. This furthermore acts to control for the effect of the power transform and additive Gaussian noise with scaling components of normality normalization, which we also showed evidence for in our earlier rebuttal comments pertaining to the motivation and selection of the hyperparameters.
> >
> > The results and analyses altogether demonstrate that normality normalization is a highly effective normalization layer across a wide range of dataset scales; this is in addition to the many useful and interesting properties of normality normalization, which we have explored throughout the paper, and have furthermore demonstrated in our previous rebuttal here – please see our rebuttal for details.
> >
> > We would be highly appreciative if you could take a moment to read through our rebuttal, and consider increasing the score for our submission.

---

> > > ### Comment · Reviewer_BP3W · 2024-12-01
> > >
> > > Thanks for your reply, I'll retain my score.

---

> ### Author Response · Authors · 2024-12-04
>
> Dear Reviewer,
>
> Based on your immediately recent score change from 6 to 5, quoting the ResNet50 experiment on ImageNet we explored during the rebuttal period, we emphasize that this was an experimental result we obtained during, and in response to, the rebuttal period under severe time constraints. As we described, the purpose of this experiment was to demonstrate the competitive performance of BatchNormalNorm vs. BatchNorm on the ImageNet dataset under the same constraints, and controlling for the same conditions, which we did successively – we were not able to train the model under the best circumstances possible.
>
> This was not included in the original submission, but again, a response to the review process and with limited time availability. Our experimental results, both in the original paper and elsewhere throughout the rebuttal period, point comprehensively to a performant normalization layer.
>
> We ask that you re-consider reverting your score to its original value. Your original score was a reflection of the original paper – this experiment, as we describe here and elsewhere, was investigated under the time constraint of the rebuttal period, and with the goal of demonstrating that on the ImageNet dataset and under the same conditions, normality normalization remains a competitive normalization layer – and we were successful in doing so.

---

### Official Review · Reviewer_buKH · 2024-10-28

**Soundness:** 3
**Presentation:** 3
**Contribution:** 2
**Rating:** 6
**Confidence:** 2

**Summary:**

The authors propose "normality normalization," a new layer that promotes normal distribution properties in neural network features by using a power transform and Gaussian noise. They back their method with experiments showing improved generalization and robustness compared to traditional normalization techniques. This approach performs well across various model architectures and increases resilience to random perturbations, offering a potentially valuable alternative for stabilizing deep network training.

**Strengths:**

* $\textbf{Theoretical Foundation}$: The use of information theory to support Normality Normalization is robust and well-articulated, highlighting benefits for maximizing representation capacity and robustness.

* $\textbf{Effective Generalization Result}$: Experimental results across multiple architectures and datasets demonstrate consistent improvements in model generalization.

* $\textbf{Comprehensive Analysis}$: The paper provides a detailed explanation of the power transform method, parameter estimation, and noise robustness, making the approach well-documented and technically thorough.

**Weaknesses:**

* $\textbf{Limited Baseline Comparisons}$: Comparisons focus on BatchNorm and LayerNorm but do not include other normalization methods like GroupNorm or adaptive techniques, which weakens the generalizability of claims.


* $\textbf{Lack of Practical Efficiency Metrics}$: The paper does not address the computational cost, making it hard to evaluate whether benefits outweigh added complexity in real-world applications.

**Questions:**

Can the approach generalize to other tasks, such as unsupervised learning, where feature distribution is critical?

---

> ### Author Response · Authors · 2024-11-22
>
> To our dear Reviewer,
>
> We address each of your comments below:
>
> >
> >"Comparisons focus on BatchNorm and LayerNorm but do not include other normalization methods like GroupNorm or adaptive techniques, which weakens the generalizability of claims."
> >
>
> We would kindly like to point out that in Subsection 5.3 Effectiveness Across Normalization Layers and via Figure 1, we do demonstrate experimentally the effectiveness of normality normalization when it is used to augment other normalization techniques, including both GroupNorm and InstanceNorm.
>
> Furthermore, please note that our experiments with vision transformers (ViTs) in Table 2 does use adaptive techniques, by means of the AdamW optimization algorithm; please refer to Appendix Subsection C.2 for details on how this adaptive optimization algorithm was used.

---

> ### Author Response · Authors · 2024-11-22
>
> >
> >"The paper does not address the computational cost, making it hard to evaluate whether benefits outweigh added complexity in real-world applications."
> >
>
> We have now included in Appendix D.6 Speed Benchmarks, a comparison between the running times of BatchNormalNorm and BatchNorm. The plots shows a close correspondence for test-time performance, with a larger deviation at training time. However, it is worth noting that the operations performed in BatchNormalNorm do not benefit from the low-level optimizations in modern deep learning libraries, afforded to the constituent operations of BatchNorm. Furthermore, the present work serves as a foundation, both conceptual and methodological, for future works which may continue leveraging the benefits of gaussianizing. We believe improvements to the runtime of normality normalization can be obtained in future work, by leveraging approximations to the operations performed in the present form of normality normalization, or by leveraging low-level optimizations.

---

> > ### Author Response · Authors · 2024-11-22
> >
> > Furthermore, we have made the following very valuable additions to the paper:
> >
> > 1. We have added a new section Appendix D.1 Other Noise-Based Techniques, where we investigate how differing noise techniques such as Gaussian dropout, compare to our proposed method of additive Gaussian noise with scaling, and for which the results are shown in Figure 7. We demonstrate that our proposed noising method works better. We also show our method works best when $s$ is set according to the minibatch statistics, i.e. not as a fixed constant, which adds further novelty and value to the method. The discussion contained in Appendix D.1 is also of interest.
> > 1. We have added a new section Appendix D.4 Effect of Degree of Gaussianization, which explores how the extent of the gaussianization relates to model performance via Figure 10, demonstrating that increasing gaussianity does improve performance,
> > 1. We have added a new section Appendix D.8 Uncorrelatedness, Joint Normality, and Independence Between Features, which demonstrates via Figure 14 the increased joint gaussianity normality normalization imbues, the resulting reduced correlation between channels of the same layer, and the increased extent of independence between channels of the same layer, the latter of which has previously been shown to be beneficial in neural networks, as we describe in Subsection 2.3,
> > 1. We have a new motivation in Subsection 2.3 Maximally Independent Representations which explores feature correlation, joint normality, and independence, between channels in the context of gaussianization, citing why increased independence can be valuable in learning models,
> > 1. We have added a new section Appendix D.3 Experiments with Data Augmentations, where via Table 4 we demonstrate the improvement in performance that can be leveraged by employing commonly used techniques such as data augmentations, whilst still demonstrating that the models trained with LNN perform better than those trained with LN,
> > 1. We have added a new section Appendix D.2 Controlling for the Power Transform and the Additive Noise, where through Figure 8 we demonstrate the effect each component of the normalization layer has. We controlled for the effect of the power transform by setting $\xi$ to $0$ – these models are denoted by "BNN w/o noise". The results demonstrate a clear benefit from both components of the normalization layer; the power transform, and the additive Gaussian noise with scaling,
> > 1. We have added a new section Appendix D.7 Normality at Initialization, demonstrating via Figure 13 that at initialization, both BatchNormalNorm and BatchNorm exhibit gaussianity; but that via Figure 5, only BatchNormalNorm enforces and maintains this gaussianity through training,
> > 1. We have added a new section Appendix D.5 Training Convergence, demonstrating via Figure 11 that the general trends in training and validation curves remain similar when using normality normalization. This is valuable because it suggests the understanding deep learning practitioners have obtained for training models with conventional normalization layers, remains applicable when augmenting those normalization layers using normality normalization,
> > 1. We have added a new paragraph in Section 6 Related Work & Future Directions: Gaussianization, regarding other gaussianization techniques which may be of interest for future work,
> > 1. In the introduction we have added further motivation for gaussianity in paragraph 3, through the perspective of neural networks as gaussian processes,
> > 1. We have changed the use of the term standardization, to align more closely with the deep learning literature, which conventionally uses the term normalization. This was done to avoid the possibility of confusing the reader – for this reason we have also changed the paper title,
> > 1. We have made several improvements throughout the text.
> >
> > We'd really like to thank you for your time and consideration – your review has helped further strengthen the work.
> >
> > We have sincerely made every attempt to comprehensively and concretely address each of your comments; through the added experiments, the additional analyses, and the refinements made to the paper. Additionally, we have made several further improvements to the work, which we listed here.
> >
> > Given this, we sincerely ask that you consider increasing your score.

---

> ### Comment · Reviewer_buKH · 2024-11-24
> **Response to Rebuttal**
>
> I thank the authors for their explanations and responses. Based on the revised version of their work, I believe it deserves a change from 5 to 6.

---

> ### Author Response · Authors · 2024-12-03
>
> Dear Reviewer,
>
> Thank you for your consideration here.
>
> Since our last correspondence, we have contributed several additional insights and analyses as part of the work. Here we provide a highlight of the additional contributions:
> 1. We engaged in a rich discussion surrounding why the same value for $\xi$ transcends different models & tasks, and furthermore pointed to possible directions for future work which could explore these in full. We suggested these perspectives may contribute to fundamentally new connections and insights, between the representations of data in the form of the activations of successive layers in a neural network, and other research areas, such as those pertaining to the reliable transmission of information across a noisy medium (information & communications theory), and the recovery of a sample when observing it after it has been randomly perturbed (Bayesian inference and denoising). These discussion points can be found in full in the following comments: https://openreview.net/forum?id=9ut3QBscB0&noteId=DdOHz401ma, https://openreview.net/forum?id=9ut3QBscB0&noteId=ivVAyK8bUI, https://openreview.net/forum?id=9ut3QBscB0&noteId=fl76He55ud . The perspectives we shared suggest that $\xi$ may have a role that is not typically associated with that of an ordinary hyperparameter, as an appropriate value may in fact be intimately tied to the properties of the normal distribution itself.
> 1. We presented a connection between mean-squared error estimation, as demonstrated by our relative error measure in Subsection 5.6 Noise Robustness Definition 1 and through the experimental results in Table 3, and the mutual information between the activations of successive model layers. This discussion helped to further cement the motivation behind the work, and can be found in full here: https://openreview.net/forum?id=9ut3QBscB0&noteId=dPi0RaIdZK .
> 1. We further demonstrated that models trained with normality normalization continue to scale with the use of additional techniques for improving generalization performance, and continue to outperform models trained with other normalization layers. This acted to further substantiate the claims made in the paper regarding the general effective of normality normalization. The new results and the surrounding discussion, can be found in this comment: https://openreview.net/forum?id=9ut3QBscB0&noteId=FSJp7er1K4 .
>
> We would therefore like to inquire if you could consider a further increase to your score for the submission, based on these highlights of the newly-made additions to the work.
>
> Once again thank you very much for your time and consideration.

---

### Official Review · Reviewer_HQv1 · 2024-10-30

**Soundness:** 3
**Presentation:** 2
**Contribution:** 4
**Rating:** 8
**Confidence:** 3

**Summary:**

The main contribution is a novel parametric layer for deep nets that improves the accuracy of image classification across models and datasets. The layer design is inspired by traditional normalization layer. Recent papers show that normalization layers (such as batch and layer normalization) make intermediate data distribution Gaussian across the layers at initialization. To go beyond initialization, this paper proposes to maintains the gaussian property during and after training using power transform (Yan&Johnson). In my opinion, the paper provides a value insights on deep learning training in addition to empirical improvements.

**Strengths:**

The main strength of the paper is its significant improvement in accuracy for image classification across various datasets. This improvement relies on a specific parametric layer that can be integrated into different neural architectures. In addition to this empirical contribution, the paper provides valuable insight into the mechanisms of normalization layers in deep learning, suggesting that making intermediate representations Gaussian enhances training.

**Weaknesses:**

The weakness is  **presentation and motivation**.
- There are several methods beyond power transforms for converting data distributions to Gaussian, including quantile transformation. I implemented quantile transformation myself, which requires no parameters like $\lambda$. However, after normalization, I observed that training became significantly slower and did not yield better generalization accuracy. Given the claim that Gaussian features improve performance, it's essential to verify if other Gaussian transformations, such as quantile transformation, also enhance performance. Since the implementation is easy, I recommend authors to conduct initial experiments on small datasets.
- In my opinion, the writing could be significantly improved. I found it challenging to connect the various concepts, such as the "best-signal" case, the mutual information framework, and noise robustness, to the proposed method.
- The introduction and abstract begin by explaining the ubiquity of the Gaussian distribution, attempting to **justify** why the proposed method performs well in practice. However, the concepts of the best-signal case or worst-case noise distribution for Gaussian do not clearly connect to the proposed normalization method.
- The contribution could be more effectively **connected to related literature**. For instance, some cited papers demonstrate that normalization layers make intermediate data representations increasingly Gaussian at initialization. Building on these findings, the designed layers could be motivated by preserving this Gaussian property throughout training.
- While the paper focuses on the distribution of individual coordinates, it is important to study how proposed method impact the **joint distribution** of data (across multiple features). Remarkably, references in *Neural Networks as Gaussian Processes* study the joint distribution of data not only a single feature, hence it is important to investigate the joint data distribution.

**Post-rebuttal:** I decided to increase my score **(from 6 to 8)** after reading authors response and checking extensive experiments in the authors response. I recommend authors to include results for other methods to impose Gaussianity, showing that some Gaussianfication method failed, as discussed with the authors.

**Questions:**

- In Figure 5, are the weights random or they are optimized? I am wonderding how the distributions look like after linear layers (not after normalization) when the weights are random. Notably, the data distribution can be gaussian after linear layers or activations while pre-activations are not gaussian.
-  Does power normalization enhance training convergence as well? The current results only demonstrate improvements in generalization, but I’m very interested in observing how both training and test accuracy evolve during training.
-  How crucial is it to optimize $\lambda$?
-  As noted, there are several transformations that convert data distributions to Gaussian. Why did you choose power transformation specifically?

---

> ### Author Response · Authors · 2024-11-22
>
> To our dear Reviewer,
>
> We address each of your comments below:
>
> >
> > "There are several methods beyond power transforms for converting data distributions to Gaussian, including quantile transformation. I implemented quantile transformation myself, which requires no parameters like $\lambda$. However, after normalization, I observed that training became significantly slower and did not yield better generalization accuracy. Given the claim that Gaussian features improve performance, it's essential to verify if other Gaussian transformations, such as quantile transformation, also enhance performance. Since the implementation is easy, I recommend authors to conduct initial experiments on small datasets."
> >
> and
> >"As noted, there are several transformations that convert data distributions to Gaussian. Why did you choose power transformation specifically?"
> >
>
> We agree this presents an interesting avenue for exploration. In fact, we did explore the quantile transformation as a method for gaussianizing early in the planning and exploration of this work; however we found it to work sub-optimally in the context of deep neural networks because neural networks are trained using gradient descent (differentiation) with backpropagation, whereas quantile transformations are non-differentiable. This makes training networks with quantile transformations non-trivial. Thus one clear advantage of using a power transform for gaussianizing, is that it has a parametric form, and thus can integrate seamlessly in neural network model training.
>
> Furthermore, in Section 6 Related Work & Future Directions, we have added a new paragraph "Gaussianization". This paragraph explores gaussianization techniques other than power transforms, and in particular points to exploring iterative gaussianization techniques as an interesting avenue for future work.

---

> > ### Comment · Reviewer_HQv1 · 2024-11-24
> > **It is good to include experiments**
> >
> > Thanks a lot for your response. I suggest to include your results for other methods and explain your intuition on why this particular method for normalization works out.

---

> ### Author Response · Authors · 2024-11-22
>
> >
> > "In my opinion, the writing could be significantly improved. I found it challenging to connect the various concepts, such as the "best-signal" case, the mutual information framework, and noise robustness, to the proposed method."
> >
>
> We have made several improvements to the text of the paper, to improve flow and readibility; with particular attention paid to the segments you mentioned here. Furthermore, we have moved the Motivation to Section 2, closer to the beginning of the paper. The purpose of doing this is to facilitate an earlier appreciation of why normality is of interest, which we believe also helps address your comment here.

---

> ### Author Response · Authors · 2024-11-22
>
> >
> >"The introduction and abstract begin by explaining the ubiquity of the Gaussian distribution, attempting to justify why the proposed method performs well in practice. However, the concepts of the best-signal case or worst-case noise distribution for Gaussian do not clearly connect to the proposed normalization method."
> >
>
> Regarding how the best-case signal and worst-case noise distribution of the mutual information game relate to the proposed method, we first give an overview of the setting in its abstract form in Subsection 2.1.1, then relate this to learning Subsection 2.1.2. Additionally, as alluded to in our previous comment, by moving the Motivation to Section 2 of the paper, we estimate that the flow and logic of the ideas will be improved. Similarly, we believe it will facilitate an appreciation on why normality is of interest earlier in the paper.
>
> The connection to the proposed normalization layer – the subject of your inquiry – is substantiated through the experiments we conduct in Subsection 5.6 Noise Robustness. There, we demonstrated that when normality normalization is employed, models are more robust to noise at test time, which is related to a tendency towards better generalization, as explored in Subsection 2.1.2. Additionally, the strong generalization performance of normality normalization throughout our experiments in Section 5 (Subsections 5.2 Generalization Performance, 5.3 Effectiveness Across Normalization Layers, and 5.4 Effectiveness Across Model Configurations), further substantiate these claims.

---

> ### Author Response · Authors · 2024-11-22
>
> >
> > "The contribution could be more effectively connected to related literature. For instance, some cited papers demonstrate that normalization layers make intermediate data representations increasingly Gaussian at initialization. Building on these findings, the designed layers could be motivated by preserving this Gaussian property throughout training."
> >
>
> We agree this is a very interesting perspective, and have thus added a new paragraph in the introduction (3rd paragraph in the updated pdf file) which gives an overview of why normality in activations may be of interest, when considering this perspective.
>
> The paragraph now reads as: "Furthermore, normality in the representations of deep neural networks imbues them with other useful properties, such as producing probabilistic predictions with calibrated uncertainty estimates (Guo et al., 2017), and making them amenable to a Bayesian interpretation (Lee et al., 2017). However, normality in the representations of neural networks is only guaranteed at initialization, and in the infinite width limit (Neal, 1996; Lee et al., 2017). This suggests that developing a method for enforcing normality throughout model training in commonly used networks is of value."
>
> Additionally, we relate this perspective through our experiments in Subsection 5.4 Effectiveness Across Model Configurations, as well as in the paragraph Neural Networks as Gaussian Processes in Section 6 Related Work & Future Directions.

---

> > ### Comment · Reviewer_HQv1 · 2024-11-24
> > **Normalization with normalization layers**
> >
> > > The paragraph now reads as:  "Furthermore, normality in the representations of deep neural networks imbues them with other useful properties, such as producing probabilistic predictions with calibrated uncertainty estimates ... "
> >
> > These results are blind to normalization layers and only holds in asymptotic regimes. However, there are results that specifically show batch normalization layers make the data representation increasingly Gaussian across the layers at initialization (see my response on There are references proves the joint data distribution ...).

---

> ### Author Response · Authors · 2024-11-22
>
> >
> > "While the paper focuses on the distribution of individual coordinates, it is important to study how proposed method impact the joint distribution of data (across multiple features). Remarkably, references in Neural Networks as Gaussian Processes study the joint distribution of data not only a single feature, hence it is important to investigate the joint data distribution."
> >
>
> We were very excited to explore this possible facet of normality normalization as well. We have now included a new motivation in Subsection 2.3 Maximally Independent Representations, which explores joint normality across the features, how this relates to the correlation between them, and ultimately how it relates to an increase in the extent of independence between them. Additionally, in Appendix D.8 Uncorrelatedness, Joint Normality, and Independence Between Features, we demonstrate via Figure 14 the increased joint gaussianity normality normalization imbues, the resulting reduced correlation between channels of the same layer, and the increased extent of independence between channels of the same layer – this last point is cited as being beneficial in neural networks, as described in Subsection 2.3.

---

> > ### Comment · Reviewer_HQv1 · 2024-11-24
> > **There are references proves the joint data distribution become normal with normalization layers**
> >
> > Among the references you cited there are results that proves the joint data distribution become normal at initialization if we use normalization layers (see Batch Normalization Orthogonalizes Representations ...). These results are purely non-asymptotic (non mean field). I am not referring to Gaussian process results that are asymptotic and blind to important role of normalization layers.

---

> ### Author Response · Authors · 2024-11-22
>
> >
> > "In Figure 5, are the weights random or they are optimized? I am wonderding how the distributions look like after linear layers (not after normalization) when the weights are random. Notably, the data distribution can be gaussian after linear layers or activations while pre-activations are not gaussian."
> >
>
> These plots correspond to the weights after they have been optimized, which highlights the gaussianizing effect of the power transform in normality normalization. We have now clarified that the plots correspond to models which have already been trained to convergence; both in the main body of the text, as well as in the Figure 5 caption.
>
> We have also added, in Appendix Subsection D.7 Normality at Initialization an experiment which addresses the subject of your inquiry regarding the behavior of these plots for models at initialization (random weights), through Figure 13. Interestingly, the plots show that at initialization, both BatchNormalNorm and BatchNorm exhibit Gaussianity; but as evidenced in Figure 5, only BatchNormalNorm enforces and maintains Gaussianity throughout training.

---

> > ### Comment · Reviewer_HQv1 · 2024-11-24
> > **BatchNormalNorm enforces and maintains Gaussianity throughout training.**
> >
> > This is a very interesting observation. I think that results of such is much more interesting for the community than maximal representation capacity and information games which are far from a rigorous argument.

---

> ### Author Response · Authors · 2024-11-22
>
> >
> > "Does power normalization enhance training convergence as well? The current results only demonstrate improvements in generalization, but I’m very interested in observing how both training and test accuracy evolve during training."
> >
>
> To address this inquiry, we have added a new section Appendix D.5: Training Convergence, demonstrating via Figure 11 that the general trends in training and validation curves remain similar when using normality normalization. This is valuable because it suggests the understanding deep learning practitioners have obtained for training models with conventional normalization layers, remains applicable when augmenting those normalization layers using normality normalization.

---

> ### Author Response · Authors · 2024-11-22
>
> >
> >"How crucial is it to optimize $\lambda$?"
> >
>
> We address this interesting inquiry through our addition of Appendix Subsection D.4 Effect of Degree of Gaussianization, which explores how the extent of the gaussianization relates to model performance via Figure 10, demonstrating that increasing gaussianity does improve performance. This further supports our claims that gaussianizing is beneficial, and complements the evidence presented throughout the paper.

---

> > ### Author Response · Authors · 2024-11-22
> >
> > Furthermore, we have made the following very valuable additions to the paper:
> >
> > 1. We have added a new section Appendix D.1 Other Noise-Based Techniques, where we investigate how differing noise techniques such as Gaussian dropout, compare to our proposed method of additive Gaussian noise with scaling, and for which the results are shown in Figure 7. We demonstrate that our proposed noising method works better. We also show our method works best when $s$ is set according to the minibatch statistics, i.e. not as a fixed constant, which adds further novelty and value to the method. The discussion contained in Appendix D.1 is also of interest.
> > 1. We have added a new section Appendix D.3 Experiments with Data Augmentations, where via Table 4 we demonstrate the improvement in performance that can be leveraged by employing commonly used techniques such as data augmentations, whilst still demonstrating that the models trained with LNN perform better than those trained with LN,
> > 1. We have added a new section Appendix D.2 Controlling for the Power Transform and the Additive Noise, where through Figure 8 we demonstrate the effect each component of the normalization layer has. We controlled for the effect of the power transform by setting $\xi$ to $0$ – these models are denoted by "BNN w/o noise". The results demonstrate a clear benefit from both components of the normalization layer; the power transform, and the additive Gaussian noise with scaling,
> > 1. We have changed the use of the term standardization, to align more closely with the deep learning literature, which conventionally uses the term normalization. This was done to avoid the possibility of confusing the reader – for this reason we have also changed the paper title,
> > 1. We have made several improvements throughout the text.
> >
> > We'd really like to thank you for your time and consideration – your review has helped further strengthen the work.
> >
> > We have sincerely made every attempt to comprehensively and concretely address each of your comments; through the added experiments, the additional analyses, and the refinements made to the paper. Additionally, we have made several further improvements to the work, which we listed here.
> >
> > Given this, we sincerely ask that you consider increasing your score.

---

> ### Comment · Reviewer_HQv1 · 2024-11-25
> **Very interesting result, but the presentation can be improved a lot**
>
> I still vote for the acceptance with score 6. I can not increase my score since the presentation is fragments of disconnected pieces of studies on Gaussian distribution. Authors could connect their results to diffusion processes or variational auto-encoders instead of Mutual Information Game. They could talk about information bottleneck principle which is much more interesting and related than maximum compact presentations. I believe that if this result was presented well, could be much more impactful.
>
> Since I really like the results, I would like to give some hints that maybe helpful to improve the writing. Generally, I think that the result should be presented as novel discovery not explaining why Gaussianifcation helps. We do not know why Gaussianifcation helps and provided intuitions can not explain the reason why. What is important to deliver the main finding "Gaussianifcation helps if it is incorporated in neural networks". This is an original exciting result.  If I wanted to present this result, I would present it in the following way:
> - Start intro with inspirations from diffusion process and variational auto encoders that want to achieve latent Gaussian distributions.
> - Inspire the question what happens if the data representation is enforced to be Gaussian
> - Showing experimental results at intro that normalization improve the performance
> - A section on normalization techniques explored and show positive and negative results for methods that are effective and methods that are not effective
> - A section on related results about normalization layers (batch norm and layer norm) and show they also make data Gaussian at initialization.
> - A detailed discussion section that provide intuitions related to mutual information games (current discussions in section 2), or possibly information bottleneck principle.

---

> ### Author Response · Authors · 2024-12-01
>
> Dear Reviewer,
>
> We had indeed accounted for this perspective previously, since in our Related Work & Future Directions section we had previously written "Given its gaussianizing effect, layers trained with normality normalization may be amenable to a non-asymptotic approximation to Gaussian processes, which prior works have investigated in the context of batch normalization (Daneshmand et al., 2021).".
>
> However, we appreciate that because the 3rd paragraph of the introduction did not comment on this, that it may not be immediately clear to readers that the connection to the non-asymptotic aspect of normalization layers is being made later in the Related Work & Future Directions.
>
> Therefore, we have decided to unify the discussion around neural networks as gaussian processes, together with the non-asymptotic perspective, in the Related Works & Future Directions section.
>
> To further complement this discussion, we have now included a reference to a work which sought to help address the disparity between the mean field and finite width analyses of neural networks at initialization.
>
> We have also reinforced the idea that normality normalization enforces and maintains normality – throughout training; which further complements the discussion.
>
> The complete paragraph in the Related Work & Future Directions Section now reads as follows:
>
> "Neal (1996) showed that in the limit of infinite width, a single layer neural network at initialization approximates a Gaussian process. This result has been extended to the multi-layer setting by (Lee et al., 2017), and Jacot et al. (2018); Lee et al. (2019) suggest the Gaussian process approximation may remain valid beyond network initialization. However, these analyses still necessitate the infinite width limit assumption.
>
> Recent work has shown that batch normalization lends itself to a non-asymptotic approximation to normality throughout the layers of neural networks at initialization (Daneshmand et al., 2021). Given its gaussianizing effect, layers trained with normality normalization may be amenable to a non-asymptotic approximation to Gaussian processes – throughout training. This could help to further address the disparity in the analysis of neural networks in the infinite width limit, for example as in mean-field theory, with the finite width setting (Joudaki et al., 2023)."
>
> Consolidating these topics by placing them in the same section makes the discussion more valuable, and the most sensible place for this is in the Related Work & Future Directions section.

---

> ### Author Response · Authors · 2024-12-01
>
> We next describe how we have addressed your uncertainty about the motivation.
>
> Earlier you had pointed out that you found it challenging to "connect the various concepts, such as the "best-signal" case, the mutual information framework, and noise robustness, to the proposed method." You had also mentioned "the concepts of the best-signal case or worst-case noise distribution for Gaussian do not clearly connect to the proposed normalization method." We had initially addressed this in our previous rebuttal comment linked here https://openreview.net/forum?id=9ut3QBscB0&noteId=5yIcAY6CTo .
>
> However, we reflected further on what the source of the challenge was likely to be, and what follows next addresses it more precisely.
>
> First, recall we had originally cited the mutual information game and noise robustness as motivation for using the normal distribution to encode activations, and for using the normal distribution to randomly perturb activations. Recall also we mentioned in our rebuttal comment that "The connection to the proposed normalization layer – the subject of your inquiry – is substantiated through the experiments we conduct in Subsection 5.6 Noise Robustness. There, we demonstrated that when normality normalization is employed, models are more robust to noise at test time, which is related to a tendency towards better generalization, as explored in Subsection 2.1.2."
>
> After having reflected further on this, we believe that challenges in making these connections, would likely be caused by it not being sufficiently/immediately clear how encoding activations using the normal distribution (part of the mutual information game motivation) is related – operationally – to noise robustness, in particular as it pertains to the results we presented in Subsection 5.6 Noise Robustness and via Table 3.
>
> These connections are made clearer by the next paragraph, which connects the concepts of mean-squared error, when recovering a signal after it has been perturbed by noise (which is analogous to what we measure in Subsection 5.6 Noise Robustness via Table 3), with mutual information (which together with noise robustness, formed the motivation in Subsection 2.1 Mutual Information Game & Noise Robustness):
>
> Added Paragraph:
>
> "Finally, there exists a close correspondence between the mutual information between the input and the output of a channel subject to additive Gaussian noise, and the minimum mean-squared error (MMSE) in estimating (or recovering) the input given the output (Guo et al., 2005). This suggests that when Gaussian noise is added to a given layer’s activations, quantifying the attenuation of the noise across the subsequent layers of the neural network, as measured by the mean-squared error (MSE) relative to the unperturbed activations, provides a direct and measurable proxy for the mutual information between the activations of successive model layers. This latter perspective is also of interest in the information bottleneck method (Tishby & Zaslavsky, 2015), which is interested in quantifying the mutual information between neural network layers."
>
> Added References:
>
> Guo et al., 2005, Mutual information and minimum mean-square error in gaussian channels, ieeexplore.ieee.org/document/1412024
>
> Tishby & Zaslavsky, 2015, Deep learning and the information bottleneck principle ieeexplore.ieee.org/document/7133169
>
> We are including this paragraph in the presumed camera-ready version of the paper; as a 3rd paragraph in Subsection 2.1.2 Relation to Learning.
>
> This paragraph clarifies and substantiates a profound connection; between mean-squared error estimation, as demonstrated by our relative error measure in Subsection 5.6 Noise Robustness Definition 1 and through the experimental results in Table 3, and the mutual information between the activations of successive model layers. This also addresses your inquiry about making other possible connections, such as with the information bottleneck method.
>
> Our current proposed changes serve to close the loop on the logic of the motivation, through the additional segments presented, which as mentioned will be included in the presumed camera-ready version of the paper. This comes in addition to the new connections made, and the recent changes which have already been incorporated in the up-to-date submission. Furthermore, the contents of the paper and our rebuttals address, together with other items, your comments.
>
> Given these elements, together in context of the comprehensive additional experiments and analyses we have added based on your review, and additionally in context of your continued enthusiasm for the work, we sincerely ask that you consider increasing your score for our submission.

---

> > ### Comment · Reviewer_HQv1 · 2024-12-03
> > **Increased score**
> >
> > I appreciate your time for explaining the connection to mutual information game and also other related topics. I increased my score since I believe the result is original and very interesting.

---

### Official Review · Reviewer_G4ZL · 2024-10-31

**Soundness:** 4
**Presentation:** 3
**Contribution:** 3
**Rating:** 8
**Confidence:** 4

**Summary:**

This paper proposes a novel normalization layer, "normality normalization", which aims to achieve almost-Gaussian pre-activations, which goes one step further than any previous normalization layer in ensuring Gaussianity of pre-activations.  They motivate this by citing the intension behind design of previous normalization techniques, information-theoretic arguments that Gaussian distribution has the highest capacity, is more noise-robust, and it simplifies dependencies to correlations, and cite parts of literature that assume or approximate Gaussianity as a desirable property.

The key technical idea is to use Yeo-Johnson power transform, which aims to make the distribution symmetric and make the tails more like a Gaussian distribution's tails, via a parameter $\lambda$. The best parameter $\hat \lambda$  can be tuned via a maximum likelihood estimation (MLE) loss that captures the normality of the values, given the mean and std of the transformed values. However, because there is no closed form solution for this, this would require an iterative approach. The authors argue that a second-order approximation of this MLE  is sufficient for sufficiently accurate estimate of $\hat\lambda,$ which allows them to compute it via a single Newton iteration.

They show that empirically, this novel normalization leads to higher validation accuracies across the board, for Layer, Instance, and Batch Normalization,  in ResNet, and ViT architectures. They further show that this improved accuracies hold accross various depths, widths, suggesting that it is not an ad hoc or highly sensitive behavior, but rather a robust improvement across the board.

**Strengths:**

I very much like overall message and contribution of this paper. First, authors present a a very well motivated argument for why Gaussianity is a good objective in neural networks. This is substantiated by several information-theoretic arguments, as well as the recognition that achieving Gaussianity has been the motivation behind other important modules, such as batch normalizatioin. After recognizing Gaussianity as a key objective in neural architecture design, they set out to solve it in a more principled manner. Furthermore, while the straightforward iterative solution to the power transformation might be too costly, they find that a quadratic approximation is good enough approximation, which can be solved in a single step, adding further novelty and value to their contribution. Finally, the empirical results make a very solid case for the empirical value of this new type of normalization layer.

**Weaknesses:**

Perhaps the main weakness that I currently perceive is how solid is the empirical case that the paper currently presents. Let me try to break this down:
- Given that noise factor $\xi$ is a hyper parameter, it would be nice to have a plot that shows the accuracy for various values of it. This would be quite important to assess the empirical vaulue of the results. Is a wide range of values for $\xi$ good enough? Or it requires a rather careful tweaking, which is problem/model-dependent? If it is the latter, it might also warrant doing a nested cross validation, (a separate validation and test set, to pick the best value of $\xi$)
- Perhaps the authors can report the percentage increase of time for training networks with BNN or LNN compared to those with BN/LN, so that readers have a better idea of what type of time-accuracy trade off their method offers? While the authors argue that the complexity of this new layer is $O(D)$, the hidden constants here might be important and not negligible. For example, compared to a classical BN or LN layer, there might be 5x or 10x more computations, which is good to report.
- I think adding a few or at least one comparable method to the empirical results is certainly helpful to readers.  While I recognize that there might not be a comparable method in the sense that they don't modify normalization layers, is it possible to compare to methods that involve injecting noise for stability or improving accuracy?
- On a related note, someone might argue that empirical value from the method is solely due to the noise injection, and not the particular power transform method. Thus, perhaps a few more empirical tests (adding noise to classic BN or LN, or an existing method that does that), would help the reader to assess the value of this work a lot!

While I think the writing of the paper is reasonably good, there are a few things that can be improved.
- Earlier in the text, namely in section 2 and introducing $\mathbf{h} = (h_i)_{i=1}^N$,  it was not clear to me what does $N$ represent. While it became clear later that this could be either the batch-wise dimension or across the feature dimension or channels, it wasn't mentioned earlier in the text. This also made it confusing of what type of normality the authors are proposing (across batch or features). Perhaps some clarifying sentences could help the readers and avoid their confusing!
- Authors use "standardization" to refer to the mean-reduction/std-division step (for example in Algorithm 1). I think this terminology is rather confusing, because standardization typically refers to this when it is done as a pre-processing step (just one before training), and not when it is part of the model and applied during training. In general, please try to avoid terminology that conflicts with existing ones.
- While technically speaking, there is nothing wrong with the title, IMHO, the current title lacking a bit awkwardly worded, which might give the wrong first impression to some readers (or cause them to not read it at all). If I may make a suggestion, a simple and descriptive title might do the paper more justice and avoid bad first impressions.

**Questions:**

- I understand the idea behind introduction of $s$ in the Gaussian noise is to make the scale of noise comparable to scale of the pre-activations. But if that's the goal, why not use a multiplicative noise that automatically achieves it?
- Just out of curiosity, Suppose we we have matrix $X$ which is $n\times d$ where $d$ is feature dimension and $n$ is batch size. Now, suppose we do BNN across the batch and achieve semi-Gaussian pre-activations. What happens to the distribution of pre-activations across the feature dimension? In other words, I wonder what are the effeects of normality normalization on the dimension that it is not explicitly normalizing.

**Details Of Ethics Concerns:**

No concerns

---

> ### Author Response · Authors · 2024-11-22
>
> To our dear Reviewer,
>
> We address each of your comments below:
>
> >
> > "someone might argue that empirical value from the method is solely due to the noise injection, and not the particular power transform method. Thus, perhaps a few more empirical tests (adding noise to classic BN or LN, or an existing method that does that), would help the reader to assess the value of this work a lot!"
> >
>
> This is a great point, which we have addressed by adding a new section Appendix D.2 Controlling for the Power Transform and Additive Noise, where through Figure 8 we conducted additional experiments demonstrating the effect each component of the normalization layer has. We controlled for the effect of the power transform by setting $\xi$ to $0$; these models are denoted by "BNN w/o noise". The results demonstrate a clear benefit from both components of the normalization layer; the power transform, and the additive Gaussian noise with scaling.
>
> Regarding the sensitivity to the free parameter $\xi$, we set this to a single value for the two types of architectures we used (ResNet/WideResNet and ViT). We demonstrated that with this single value, models performed well across the board; despite changes in dataset, architecture size (depth/width), and minibatch size.
>
> To address your question of how this value for $\xi$ was chosen, as we describe in Appendix Subsections C.1 and C.2, these were chosen solely using preliminary experiments which aimed to evaluate, at what point further increases to $\xi$ led to unstable training behaviors. Given we used a consistent value for $\xi$ across our experiments for the ResNet/WideResNet and ViT architectures, this shows the effectiveness of the method was not sensitive to the value of $\xi$.

---

> ### Author Response · Authors · 2024-11-22
>
> >
> > "Given that noise factor $\xi$ is a hyper parameter, it would be nice to have a plot that shows the accuracy for various values of it. This would be quite important to assess the empirical vaulue of the results. Is a wide range of values for $\xi$ good enough? Or it requires a rather careful tweaking, which is problem/model-dependent? If it is the latter, it might also warrant doing a nested cross validation, (a separate validation and test set, to pick the best value of $\xi$)"
> >
>
> Thank you for this excellent suggestion, we have now added precisely this experiment in Appendix D.2 Controlling for the Power Transform and Additive Noise via Figure 9, which also serves to demonstrate that the previously chosen value of $\xi$ in BatchNormalNorm, works consistently well across the model and dataset combinations.

---

> > ### Comment · Reviewer_G4ZL · 2024-11-23
> >
> > In Figure 9, there seems to be two main messages: one is that the value of $\xi$ does have a big impact on the accuracies, and second is that the optimal values transfer between models and tasks.
> >
> >
> > Question: If the accuracy is sensitive to value of $\xi$, why a generic optimal value that transcends tasks and models should exist?
> >
> > Right now it's slightly difficult to answer this, because authors haven't plotted values above $0.4$ in Figure 9.  Assuming these are validation/test scores, one would normally expect a peaked behavior there,  and it would be nice to see it. Namely, if the peaks for different tasks/model are shared/different.
> >
> > If different models & tasks do share a common optimal $\xi$, can authors explain/justify this?

---

> > > ### Author Response · Authors · 2024-11-26
> > >
> > > Dear Reviewer,
> > >
> > > We address these very interesting questions next – we demonstrate that the implications, and possible avenues for future work, are incredibly tantalizing.
> > >
> > > First, we will comment on our experimental experience as it pertains to how $\xi$ affects performance. Throughout all of our many, many experiments involving BatchNormalNorm (LayerNormalNorm discussed subsequently), we observed the following: not in a single instance, over the numerous experiments we ran, did $\xi=0.4$ lead to unstable training behavior – it is this absolute reliability of $\xi=0.4$ that made it the upper bound in all of our experiments. In contrast, and most pertinent to the question you have raised here, is that although $\xi \ge 0.6$ (for example) may work in some experiments, it is somewhat dependent on the experimental setup and the random seed - and thus not guaranteed to be performant or work at all. Therefore in many of our experiments, values of $\xi \ge 0.6$ led to no progress in training at all. This complements our description in Appendices C.1 and C.2, in which we made note of training stability as justification for choosing a fixed value for $\xi=0.4$. To summarize, it is possible in some experimental settings that $\xi \ge 0.6$ would work well, but it does not provide the same reliability – to an absolute level – that we have found $\xi = 0.4$ to provide, which is why we set it as an upper bound in our experiments.
> > >
> > > The natural next question, as you raise as well is: why does a certain value of $\xi$ work universally, and transcend different network and dataset settings? We believe the following perspectives may be illuminating, and may demonstrate that $\xi$ may have a role that is not typically associated with that of an ordinary hyperparameter, as an appropriate value may in fact be intimately tied to the properties of the normal distribution itself:

---

> ### Author Response · Authors · 2024-11-22
>
> >
> > "Perhaps the authors can report the percentage increase of time for training networks with BNN or LNN compared to those with BN/LN, so that readers have a better idea of what type of time-accuracy trade off their method offers? While the authors argue that the complexity of this new layer is $O\left(D\right)$, the hidden constants here might be important and not negligible. For example, compared to a classical BN or LN layer, there might be 5x or 10x more computations, which is good to report."
> >
>
> We have now included in Appendix D.6 Speed Benchmarks, a comparison between the running times of models using BatchNormalNorm vs. BatchNorm. The plots shows a close correspondence for test-time performance, with a larger deviation at training time. However, it is worth noting that the operations performed in BatchNormalNorm do not benefit from the low-level optimizations in modern deep learning libraries, afforded to the constituent operations of BatchNorm. Furthermore, the present work serves as a foundation, both conceptual and methodological, for future works which may continue leveraging the benefits of gaussianizing. We believe improvements to the runtime of normality normalization can be obtained in future work, by leveraging approximations to the operations performed in the present form of normality normalization, or by leveraging low-level optimizations.

---

> ### Author Response · Authors · 2024-11-22
>
> >
> > "I think adding a few or at least one comparable method to the empirical results is certainly helpful to readers. While I recognize that there might not be a comparable method in the sense that they don't modify normalization layers, is it possible to compare to methods that involve injecting noise for stability or improving accuracy?"
> >
> and
> >
> >"I understand the idea behind introduction of $s$ in the Gaussian noise is to make the scale of noise comparable to scale of the pre-activations. But if that's the goal, why not use a multiplicative noise that automatically achieves it?"
> >
>
> We were excited by this suggestion, as it presents an opportunity to demonstrate the novelty and value of the proposed noising mechanism. First, Appendix D.1 Other Noise-Based Techniques now compares and contrasts our proposed method with Gaussian dropout, over several retention probabilities $p$, as shown in Figure 7. Here we demonstrate that our proposed noising method works better. We also show our method works best when $s$ is set according to the minibatch statistics, i.e. not as a fixed constant, which adds further novelty and value to the method.
>
> Furthermore, as we explore in Appendix D.1 Other Noise-Based Techniques, there is a significant difference between other works applying noise, such as Gaussian dropout, and the present work which uses additive Gaussian noise with scaling. Gaussian dropout scales activations multiplicatively, which has the following subtle but significant consequence: the effect and scale of the noise is incorporated directly during gradient descent via backpropagation – this boils down to the fact that multiplicative operations carry over when taking gradients. In contrast, the additive Gaussian noise is not directly incorporated into the gradient descent updates during backpropagation, because additive effects are eliminated when taking gradients. In this sense, the noise from additive Gaussian noise is "confusable", because the backward pass accounts for a different activation value than what was realized during the forward pass. This implies that models which can successfully be trained with additive Gaussian noise, should be more robust, and have better generalization – which our experiments demonstrate.

---

> ### Author Response · Authors · 2024-11-22
>
> >
> >"Earlier in the text, namely in section 2 and introducing $\mathbf{h}=\left(h_{i}\right)_{i=1}^{N}$, it was not clear to me what does $N$ represent. While it became clear later that this could be either the batch-wise dimension or across the feature dimension or channels, it wasn't mentioned earlier in the text. This also made it confusing of what type of normality the authors are proposing (across batch or features). Perhaps some clarifying sentences could help the readers and avoid their confusing!"
> >
>
> Thank you very much for this suggestion - we agree that relating $N$ to the normalization layer setting earlier in the text would be helpful. Please now see Section 3 Background: Power Transform, where we have there taken the opportunity to clarify what $N$ corresponds to, in the second paragraph.

---

> ### Author Response · Authors · 2024-11-22
>
> >
> > "Authors use "standardization" to refer to the mean-reduction/std-division step (for example in Algorithm 1). I think this terminology is rather confusing, because standardization typically refers to this when it is done as a pre-processing step (just one before training), and not when it is part of the model and applied during training. In general, please try to avoid terminology that conflicts with existing ones."
> >
>
> We really appreciate this point being raised – we acknowledge that although standardization is technically a correct term for describing mean subtraction and diving the result by the standard deviation, in the context of the deep learning literature it may contribute to confusion, because here normalization is the term which is normally used, and not standardization. Therefore we have modified the text to reflect this terminology better, including changing the title, for the very purpose of avoiding this possible source of confusion.

---

> ### Author Response · Authors · 2024-11-22
>
> >
> > "Just out of curiosity, Suppose we we have matrix $X$ which is $n \times d$ where $d$ is feature dimension and $n$ is batch size. Now, suppose we do BNN across the batch and achieve semi-Gaussian pre-activations. What happens to the distribution of pre-activations across the feature dimension? In other words, I wonder what are the effeects of normality normalization on the dimension that it is not explicitly normalizing."
> >
>
> We believe this question is very interesting, and believe it equivocates to the following formulation in the case of BatchNormalNorm: when one dimension is being explicitly gaussianized (a given channel's activations across the minibatch entries), what happens across the alternative dimension (ex: the distribution of activations across the set of channels in the layer). This is therefore equivalent to asking about the possibility of joint normality across a separate dimension, which is very interesting to consider!
>
> This inquiry led us to include a new motivation in Subsection 2.3 Maximally Independent Representations, which explores joint normality across the features, how this relates to the correlation between them, and ultimately how it relates to an increase in the extent of independence between them. Additionally, in Appendix D.8 Uncorrelatedness, Joint Normality, and Independence Between Features, we demonstrate via Figure 14 the increased joint gaussianity normality normalization imbues, the resulting reduced correlation between channels of the same layer, and the increased extent of independence between channels of the same layer, the latter of which is shown to be beneficial in neural networks, as described in Subsection 2.3.

---

> > ### Author Response · Authors · 2024-11-22
> >
> > Furthermore, we have made the following very valuable additions to the paper:
> >
> > 1. We have added a new section Appendix D.4 Effect of Degree of Gaussianization, which explores how the extent of the gaussianization relates to model performance via Figure 10, demonstrating that increasing gaussianity does improve performance,
> > 1. We have added a new section Appendix D.3 Experiments with Data Augmentations, where via Table 4 we demonstrate the improvement in performance that can be leveraged by employing commonly used techniques such as data augmentations, whilst still demonstrating that the models trained with LNN perform better than those trained with LN,
> > 1. We have added a new section Appendix D.7 Normality at Initialization, demonstrating via Figure 13 that at initialization, both BatchNormalNorm and BatchNorm exhibit gaussianity; but that via Figure 5, only BatchNormalNorm enforces and maintains this gaussianity through training,
> > 1. We have added a new section Appendix D.5 Training Convergence, demonstrating via Figure 11 that the general trends in training and validation curves remain similar when using normality normalization. This is valuable because it suggests the understanding deep learning practitioners have obtained for training models with conventional normalization layers, remains applicable when augmenting those normalization layers using normality normalization,
> > 1. We have added a new paragraph in Section 6 Related Work & Future Directions: Gaussianization, regarding other gaussianization techniques which may be of interest for future work,
> > 1. In the introduction we have added further motivation for gaussianity in paragraph 3, through the perspective of neural networks as gaussian processes,
> > 1. We have made several improvements throughout the text.
> >
> > We'd really like to thank you for your time and consideration – your review has helped further strengthen the work.
> >
> > We have sincerely made every attempt to comprehensively and concretely address each of your comments; through the added experiments, the additional analyses, and the refinements made to the paper. Additionally, we have made several further improvements to the work, which we listed here.
> >
> > Given this, we sincerely ask that you consider increasing your score.

---

> ### Comment · Reviewer_G4ZL · 2024-11-22
> **very interesting work**
>
> General comment: I thank the authors for their careful reading of the reviews and their clear and on-point responses. I also congratulate them on writing a very thoughtful and interesting (and hopefully impactful) paper. As is clear in my original review, I had very positive opinions of the work from the beginning, and the changes made have certainly improved the paper quite a lot.
>
> That said, I find the comments by reviewer `hs2Z` on the lack of comprehensive experiments to be quite valid, which align with some of my earlier comments. In particular, the fact that baselines reported in the paper are lower than those reported elsewhere, suggest baselines are left (perhaps unintentionally) under-optimized. I still find the novel and interesting, and consider the evidence presented here  to be a "proof of concept" of the idea, and would hopefully lead to subsequent works that will shed more light on its practical benefits.
>
> All being considered, I will maintain my originally high score (6).
>
> One specific note: I find the answer on quasi-independence  due to normality normalizing quite interesting. I believe there are many more interesting questions on this, worth exploring (in this paper or future works)
> -  if normalizing across one dimension transfers approximately to another, you can also explore venues for applications. Namely, if one dimension is smaller, translates calculating fewer statistics and faster training time.
> - Another aspect of this is simplicity of training. For example, because LayerNorm doesn't have batch-wise dependencies, it is much easier to distribute it across GPUs or multiple nodes than BatchNorm, which at every layer, it would require gathering batch-wise statistics. I believe much of the reason that LayerNorm has grown in popularity are these practical advantages over BN. So, if LNN can remain as simple as LN, but as effective and powerful as BN, that would be a very interesting outcome. I believe this might be worth studying in a new inquiry.

---

> > ### Author Response · Authors · 2024-11-23
> >
> > Dear Reviewer,
> >
> > We appreciate your comment regarding the comprehensiveness of the experiments. However, we would like to express that we have addressed these concerns throughout the rebuttal, as we describe next.
> >
> > Regarding the scale of the experiments, in Table 2 we do have ImageNet experiments in the form of the ImageNet100 experiments, which convincingly demonstrate the superior performance of the vision transformer (ViT) trained with layer normality normalization (LNN) compared to the ViT trained with layer normalization (LN). This portion of Table 2 is given by:
> >
> > |Dataset|LN|LNN|
> > |----------|----------|----------|
> > |ImageNet100 Top1|50.78 $\pm$ 0.33|**62.39 $\pm$ 0.68**|
> > |ImageNet100 Top5|75.45 $\pm$ 0.50|**84.03 $\pm$ 0.42**|
> >
> > and demonstrates a clear improvement at a high accuracy level in context of the the experimental setup.
> >
> > Furthermore, we have been extremely comprehensive in ensuring rigor and precision in the results we report. All of our experiments – that is, every single value reported numerically or graphically – is reported across an average of $M=6$ models, each having been trained with differing random initializations for the model parameters. Thus we have ensured a high degree of confidence in all of our results.
> >
> > Regarding the baseline performances, we completely understand your perspective, and would like to clarify why the baseline performances were lower in part: we did not use augmentations. We agree that demonstrating our proposed normalization layer improves in performance with commonly used techniques, is important.
> >
> > Therefore we added a new set of experiments in Appendix D.3 Experiments with Data Augmentations via Table 4, which demonstrates the improvement in performance that can be leveraged by employing commonly used techniques such as data augmentations, whilst still demonstrating that the models trained with LNN perform better than those trained with LN. Table 4 is given as follows:
> >
> > |Dataset|LN|LNN|
> > |----------|----------|----------|
> > |SVHN|94.46 $\pm$ 0.33|**95.94 $\pm$ 0.18**|
> > |CIFAR10|73.71 $\pm$ 0.42|**75.47 $\pm$ 0.49**|
> > |CIFAR100|49.56 $\pm$ 0.42|**52.89 $\pm$ 0.51**|
> > |Food101|55.43 $\pm$ 0.57|**63.04 $\pm$ 0.72**|
> >
> > This demonstrates a significant improvement from the results in Table 2, which is facilitated through the commonly used technique of data augmentations. This demonstrates that the improvements for models trained with LNN continue to scale with the use of such techniques.
> >
> > Given this and the additional evidence we provide regarding the improved performance of our models when commonly used techniques such as data augmentations are employed, we sincerely ask that you consider increasing your score in light of this evidence.

---

> > > ### Comment · Reviewer_G4ZL · 2024-11-23
> > >
> > > I appreciate this clarification by the authors about the baselines. This does indeed answer the primary concern about the experimental evidence.

---

> ### Author Response · Authors · 2024-11-26
>
> - First, consider the mutual information between $X$ and $Y = X + Z$, for Gaussian signal $X$ and Gaussian noise $Z$, which is given by $\frac{1}{2}\log\left(1 + \frac{\sigma_{X}^{2}}{\sigma_{Z}^{2}}\right)$ – this is actually the channel capacity of the Gaussian channel in information theory. As can be seen from this expression, it is determined by the ratio of the signal and noise variances, i.e. $\frac{\sigma_{X}^{2}}{\sigma_{Z}^{2}}$, which is a function of $\xi$ (through the term $\sigma_{Z}^{2}$ which absorbs it). In communications theory, a significant amount of work has been dedicated to studying the properties of the Gaussian channel, and how systems behave for varying channel capacities. These insights may be pertinent for neural networks trained with normality normalization and provide insight into what values of $\xi$ are tolerable; ultimately whether we are investigating the properties of the normal distribution from an information & communications theory perspective, or in the setting of a neural network trained with normality normalization, they are both concerned with the reliable propagation of information – through the activations of successive layer in neural networks, and through a noisy communication channel in information theory. Furthermore, because the normalization layer acts at a unit/channel level, the value of $\xi$ which works best for a given unit/channel in one architecture, should be effectively equivalent or at least similar, to that of a unit/channel in a different architecture; ultimately in both cases we gaussianize a set of activations, an operation which in isolation is somewhat removed from the other operations which make up the network. This very much aligns with the contemporary "block-by-block" design of deep learning systems.
> - Next, consider the following alternative but perhaps equally insightful perspective: suppose there is a data point $x$ sampled from a Gaussian random variable $X$ with variance $\sigma_{X}^{2}$, to which we apply additive noise $z$ sampled from a Gaussian random variable $Z$ with variance $\sigma_{Z}^{2}$; thus we observe $y = x + z$. Consider the following: what is the relationship between the extent of additive random noise $z$ that can be applied to $x$ (which gives rise to the observed $y = x + z$), and our ability to recover the original value of $x$ (with a given level of precision/tolerance)? The relation to neural networks trained with normality normalization is as follows: if the purpose of adding noise during training is to make the neural network robust to random perturbations to its activations, for what range of $\xi$ values does the noise become too great and thus results in too much corruption of the activation value, and for which values of $\xi$ does the perturbation instead act as a regularizer and lead to improved generalization, rather than significantly corrupting the activation. Furthermore, what signal distribution $X$ is most conducive to the task of signal recovery? This perspective is profoundly related to the Bayesian inference setting, including for example in the context of minimum mean-squared error (MMSE) estimation, and to the perspective of denoising in statistics, for example in the context of soft thresholding – these are very interesting perspectives which would certainly merit new works devoted to their study. We believe the present work is of fundamental importance, as a bedrock under which such subsequent works can flourish.

---

> > ### Author Response · Authors · 2024-11-26
> >
> > Finally it is interesting to reflect on how these preceding points are intimately tied to normality normalization, more-so than to other normalization layers. Observe that both of the preceding avenues of investigation have Gaussian signals, and use Gaussian noise - in fact the first discussion point above regarding channel capacity, is intimately tied to the mutual information game setting we explore in the paper. Thus in these parallel settings, it is the normal distribution which most facilitates the task of information recovery, and again the normal distribution which makes the task of information recovery the most difficult. This is intimately tied to why, as we explicitly explored in the paper, we chose the normal distribution as the distribution of choice for the activations, and again chose the normal distribution as the distribution of choice for the random perturbations, for which becoming robust to would have the strongest regularizing effect.
> >
> > We briefly comment on the differing values for $\xi$ for BatchNormalNorm and LayerNormalNorm. The fact that $\xi=1.0$ in LayerNormalNorm, suggests that sensitivity of a particular unit's value in the collective layer, as in LayerNormalNorm, may be smaller than the sensitivity of a particular data point's value in the collective minibatch, as in BatchNormalNorm. An inquiry regarding why two differing values of $\xi$ may be appropriate, one for BatchNormalNorm and another for LayerNormalNorm, can furthermore be investigated from the following perspective: the correlation structure in a set of samples affects the extent to which random additive noise can act to perturb the samples. Because BatchNormalNorm and LayerNormalNorm normalize across differing axes; minibatch samples in BatchNormalNorm, and units in LayerNormalNorm; the generally differing correlation structures in their respective samples, implies differing noise factors are appropriate in the two contexts, which is what our experimental evidence also supports.
> >
> > Ultimately, we believe these preceding perspectives may contribute to fundamentally new connections and insights, between the representations of data in the form of the activations of successive layers in a neural network, and other research areas, such as those pertaining to the reliable transmission of information across a noisy medium (information & communications theory), and the recovery of a sample when observing it after it has been randomly perturbed (Bayesian inference and denoising).
> >
> > We'd like to emphasize that although these are profoundly interesting questions, they certainly merit new works, dedicated entirely to their study and analysis; the present work acts as a bedrock for these avenues of investigation. However, your present question and the review/rebuttal process altogether, has given us an opportunity to offer a perspective on these possibly illuminating avenues for future investigation, but in the less formal (but still insightful) setting of the present comment; thank you.
> >
> > Thank you.

---

> ### Author Response · Authors · 2024-11-26
>
> In addition to our preceding sequence of comments addressing your question of why the same value of $\xi$ transcends different tasks and models, we next would like to share further experimental evidence regarding the performance of normality normalization on the ImageNet dataset in its entirety, for which we earlier dispatched an experiment involving ResNet50 models trained with either BatchNormalNorm and BatchNorm, for which we now share the results for. The results can be found in Appendix D.3 Experiments with Data Augmentations.
>
> The experiment demonstrates an improvement for the ResNet50 model trained with BatchNormalNorm. Furthermore, this experiment was run with $\xi=0$, i.e. using BatchNormalNorm without noise (BNN w/o noise), and still demonstrated an improvement over BatchNorm (BN). This further acts to control for the effect of the power transform alone; which is also a point of inquiry you had raised in your original review.
>
> After our previous rebuttal comments, including w.r.t. our addition of a new set of experiments in Appendix D.3 Experiments with Data Augmentations via Table 4, you had shared with us that this had indeed answered your primary concern about the experimental evidence. We believe the additional experiment we have shared here on the ImageNet dataset, acts to even further address any concerns regarding experimental evidence.
>
> The results and analyses altogether demonstrate that normality normalization is a highly effective normalization layer across a wide range of dataset scales; this is in addition to the many useful and interesting properties of normality normalization, which we have explored throughout the paper, and have furthermore demonstrated during the rebuttal period.
>
> In light of all of this, we would be highly appreciative if you could take a moment to consider increasing the score for our submission.

---

> > ### Comment · Reviewer_G4ZL · 2024-11-26
> >
> > I again thank the authors for the explanations and I have no pending concerns with the technical or empirical setup of the paper. I also thank them for taking time and care during the rebuttal to answer all reviewers pointedly and with diligence.
> >
> > While I already voted for the acceptance of the paper and I consider it to have a high potential to be of high impact, I still have some concerns about the presentation and writing. For example, while the current title is better than the original one, but "putting normality in normalization" might sound confusing or off-putting to some readers due to repeating the same word twice (perhaps putting Gaussianity in normalization is less ambiguous?!). Sadly these issues might impact the future readers, or wether someone would read it at all.
> >
> > But I recognize remaining issues with writing are more stylistic rather than objective, and I hope that future readers would see past the writing and see the core messages that are interesting and valuable. . Thus, I'm happy to increase my score to 8 (increased from  6)

---

> ### Author Response · Authors · 2024-11-26
>
> Dear Reviewer,
>
> Thank you very much for your consideration.
>
> We would kindly like to note, that we did indeed previously change the paper title after carefully considering your original review. The pdf title reads: Normality Normalization. We also modified the text to reflect this terminology better, for the very purpose of avoiding this possible source of confusion.
>
> We did not at that point change the OpenReview paper title, simply because we thought it might cause confusion for any of our reviewers when referring to the paper submission on the OpenReview platform. In fact we do not see an option to change the paper title on the OpenReview platform at this time.
>
> Nevertheless we agree with your original point: although standardization is technically a correct term for describing mean subtraction and dividing the result by the standard deviation, in the context of the deep learning literature it may contribute to confusion, because here normalization is the term which is conventionally used.
>
> Thank you again for your time and consideration.

---

### Meta-Review · Area_Chair_cRkL · 2024-12-10

**Metareview:**

The paper introduces a new approach named Normality Normalization that aims at making the pre-activation distributions of neural networks Gaussian. This method utilizes a power transform to enforce Gaussianity and adds Gaussian noise to enhance noise robustness. The authors demonstrated the method's effectiveness for various models and datasets, including for instance ResNet and VITs. Interestingly, the approach is reported to consistently improve over traditional normalization methods such as BatchNorm and LayerNorm. The paper clearly has some merits but also weaknesses which I summarize below.


### **Strenghts**
- Introduces a new normalization layer that extends the idea of Gaussianity in neural network activations, supported by theoretical insights.
- Demonstrates consistent improvements in generalization and noise robustness over existing normalization methods (e.g., BatchNorm, LayerNorm).
- Rebuttal effort: the authors provided extensive new experiments and revisions during the rebuttal phase, addressing many concerns raised by reviewers.


### **Weaknesses**

Two important problems were raised:

1) The existing results are compared to suspiciously low accuracy baselines.
In response, the authors said this was due to the fact they did not use data augmentation. Additional results were later provided, but some of the new improvements seem to be more minor.

2) Lack of experiments on larger datasets.
2.1) One option would be an NLP task. In that regard, I did not find the justification given by the authors to be convincing. They write "We believe the fair comparison would have been to train a large language model from scratch". However, training nanoGPT (or even a smaller version) on a medium size dataset should be doable with a few GPUs.

2.2) When experiments on large-scale data was added, it seems the improvements are relatively minor. I agree with the comments of Reviewer hs2Z who writes "the improvement is not very significant (0.13% in Top5 and 0.34% in Top1), especially given this is a single seed."

### **Decision**

Three out of the four reviewers are in favor of accepting the paper while one reviewer is strongly against accepting it. I also would like to mention that the reviewers did a good job as they provided some constructive feedback that will surely help improve the paper.

Overall, this is a rather difficult case as all the reviewers recognize the merits of the submission which is based on a novel idea (even Reviewer hs2Z writes "The idea is interesting, original, novel, and with a reasonable motivation."). However, all the reviewers expressed or recognized concerns regarding the limited scale of the experimental results.

In conclusion, the paper has some merits but I tend to agree with Reviewer hs2Z and I believe further empirical evidence has to be provided: the authors should achieve higher accuracies for the baselines (using data augmentation if necessary) and also provide results over several runs in the large-scale setting they started to investigate at the end of the rebuttal period. I therefore recommend rejection and advise the authors to resubmit to the next conference deadline with the changes mentioned above.

**Additional Comments On Reviewer Discussion:**

Lots of discussions led some authors to increase their scores but two reviewers (one in particular) expressed concerns about the experiments. After a long discussion, Reviewer hs2Z remained on the rejection side and I believe the arguments they raised are indeed valid and need to be addressed.

---

### Decision · Program_Chairs · 2025-01-22

Reject